# Factors Influencing Margin Clearance and the Number of Stages of Mohs Micrographic Surgery in Basal Cell Carcinoma: A Retrospective Chart Review

**DOI:** 10.3390/cancers16132380

**Published:** 2024-06-28

**Authors:** Vijaya T. Daniel, Vincent Azzolino, Maria Abraham, Nicholas Leonard, Kaitlin Blankenship, Karan Lal, Julie Flahive, Regina Brown, Elizabeth Tkachenko, Shereen Teymour, Abdel Kader El Tal, Bassel H. Mahmoud

**Affiliations:** 1Department of Dermatology, University of Massachusetts, Worcester, MA 01605, USA; vijaya.daniel@umassmemorial.org (V.T.D.); vincent.azzolino@umassmed.edu (V.A.); nicholas.leonard@umassmemorial.org (N.L.); redifor1@gmail.com (R.B.); 2Department of Internal Medicine, University of Maryland, Baltimore, MD 20742, USA; maria.abraham0705@gmail.com; 3Dermatology & Skin Cancer Surgery Center, Denton, TX 76210, USA; kaitlinchiara@yahoo.com; 4Affiliated Dermatology, Scottsdale, AZ 85255, USA; kdermlal@gmail.com; 5Department of Population and Quantitative Health Sciences, University of Massachusetts Medical School, Worcester, MA 01605, USA; julie.flahive@umassmed.edu; 6Yale Department of Dermatology, New Haven, CT 06520, USA; elizabeth.tkachenko@yale.edu; 7Palisades Medical Center, Hackensack University Medical Center, North Bergen, NJ 07601, USA; steymour@gmail.com; 8Dermatology Associates, Perrysburg, OH 43551, USA; abdulkadertal@gmail.com

**Keywords:** basal cell carcinoma, Mohs, surgery, outcomes, health services research, epidemiology

## Abstract

**Simple Summary:**

We aim to understand the relationship between the size of the defect to clear basal cell carcinoma (BCC) using Mohs Micrographic Surgery (MMS) and patient demographics, as well as the relationship between the number of MMS stages and BCC subtypes. In a large retrospective study involving 7651 patients with BCC who underwent MMS, our analyses demonstrated that a higher clearance margin was associated with increased age and that an increase in the number of MMS stages was associated with higher-risk BCC. By monitoring outliers in MMS practice patterns to identify financial burdens and unnecessary surgery, it may be important to further account for patient demographics as well as tumor factors.

**Abstract:**

How patient and tumor factors influence clearance margins and the number of Mohs Micrographic Surgery (MMS) stages when treating basal cell carcinoma (BCC) remains widely uncharacterized. It is important to elucidate these relationships, as surgical outcomes may be compared nationally between colleagues. Our objective is to evaluate the relationships between defect size and patient demographics, as well as between BCC subtypes and the number of MMS stages. Our second objective is to compare practice patterns and characteristics of patients requiring MMS at academic centers and private practices. A retrospective chart review was performed using data collected at academic centers (2015–2018) and private practices (2011–2018) of BCC patients older than 18 years old who underwent MMS. In total, 7651 patients with BCC requiring MMS were identified. Academic center adjusted analyses demonstrated clearance margins 0.1 mm higher for every year’s increase in age (*p* < 0.0001) and 0.25 increase in MMS stages for high-risk BCC (*p* < 0.0001). Private practice adjusted analyses demonstrated clearance margins 0.04 mm higher for every year’s increase in age (*p* < 0.0001). Clearance margins correlate with older age, and additional MMS stages correlate with high-risk BCC, suggesting the role patient and tumor factors may play in predicting tumor clearance and MMS stages.

## 1. Introduction

Basal cell carcinoma (BCC) is a slow-growing skin cancer that represents the most common malignancy in the United States today, with an incidence that is steadily rising [1]. Histologic subtypes of BCC can be separated into low-risk (superficial, nodular, and pigmented) and high-risk (morpheaform, infiltrative, micronodular, and basosquamous) [1]. While a variety of modalities exist for the treatment of BCC, Mohs Micrographic Surgery (MMS) is often selected for lesions that are large, histologically aggressive, recurrent, or located in high-risk body locations (Figure 1). The American College of Mohs Surgery (ACMS) has recently supported the tracking of the mean number of MMS stages per surgery as a quality improvement metric for Mohs surgeons. The goal of this initiative includes both exploring the distribution of practice patterns among Mohs surgeons and potentially identifying patterns of excessive MMS stage usage that could lead to suboptimal patient outcomes or increasing healthcare costs. While the mean number of MMS stages has previously been shown not to vary between surgeons at different stages in their careers [2], it was recently demonstrated that surgeon-specific variation does exist for mean MMS stages in general. This recent study also highlighted that surgeons in solo practice were more likely to display practice patterns with a higher number of MMS stages [3], but a notable limitation of the aforementioned study was the absence of any analysis of patient characteristics, lesion location, or histologic subtypes.

Few studies have explored the extent to which patient or tumor factors may influence clearance margins and the number of MMS stages when treating BCC. One specific metric that has been most closely considered is the subclinical spread of lesions treated with MMS. Subclinical spread has been defined by the number of MMS stages needed to completely excise a lesion and is typically considered extensive when at least two, but oftentimes three or more, stages are needed for complete clearance. Few small retrospective chart reviews have shown significantly higher rates of subclinical spread for infiltrative, morpheaform, and micronodular subtypes compared with nodular and superficial subtypes [4,5], and a larger study focusing only on BCC located on the head and neck showed that there was no relationship between subclinical spread and location on the H zone of the face, patient sex, or patient age [6].

We have little evidence of the relationship between the aforementioned factors, including MMS (clearance margins, number of MMS layers), BCC subtype, and location; also, the majority of these studies have been small. It is crucial that these relationships not only be clearly elucidated but also quantified in an era when surgical performance outcomes will be compared between colleagues on a national level. The primary objectives of our study were to evaluate the relationship between the size of the defect needed to clear BCC using MMS and patient demographics, as well as the relationship between the number of MMS stages to clear the BCC and BCC subtype. The secondary objective was to compare practice patterns and characteristics of patients with BCC requiring MMS at an academic center and a private practice setting.

## 2. Materials and Methods

A retrospective chart review was conducted of BCC requiring MMS at an academic center (2015–2018) and a private practice setting (2011–2018). While two separate settings were analyzed, academic center and private practice settings cannot be directly compared to each other. In both settings, MMS was performed by board-certified Moh surgeons. Patients older than 18 years old undergoing MMS for BCC were identified from chart reviews from both practice settings. Margin clearance was defined as the postoperative defect size (prior to reconstruction) minus the preoperative lesion size. High-risk BCC pathologic subtypes were defined as any combination involving a high-risk subtype including morpheaform/fibrosing/sclerosing, invasive/infiltrative, micronodular, and basosquamous. Low-risk pathologic subtypes were defined as superficial, nodular, and superficial/nodular. The outcomes were margin clearance and the number of MMS stages.

Categorical data are presented as percentage frequencies, with continuous data presented as median values and interquartile ranges. Categorical data were analyzed using the Rao–Scott chi-square test. Continuous data were analyzed using analysis of variance. Multivariable linear regression models were constructed for clearance margins and numbers of MMS stages. Statistical significance was defined as *p* < 0.05. All statistical analyses were performed using SAS 9.4 statistical software (SAS Institute, Cary, NC, USA).

## 3. Results

Overall, 4897 BCCs requiring MMS at an academic center and 2754 BCCs requiring MMS at a private practice setting were identified. Compared to the academic center, the private practice setting’s cohort had fewer males (52% vs. 56%, *p* = 0.001), required fewer MMS stages to achieve clearance (1 (interquartile range 1, 2) vs. 2 (1, 2), *p* < 0.0001), a smaller clearance margin (4 mm (2, 6) vs. 7 mm (5, 11), *p* < 0.0001), more superficial subtypes (7.7% vs. 2.4%) and more morpheaform/fibrosing/sclerosing/invasive/infiltrative or basosquamous subtypes (11% vs. 6.8%) (*p* < 0.0001) (Table 1). However, factors that may have caused the observed difference between academic center and private practice settings cannot be determined via parallel comparison. We utilized internal statistical analysis for each unique practice setting to better understand how patient demographics and tumor factors influenced practice patterns.

Unadjusted analyses from the academic center demonstrated the clearance margin was 0.12 mm higher for males compared to females (*p* < 0.0001); the clearance margin was lower for younger patients compared to patients who were 80 years and older (*p* < 0.0001); and a 0.23 increase in number of MMS stages for high-risk pathologic subtypes compared to low-risk pathologic subtypes (<0.0001). Unadjusted analysis from the private practice setting showed that the clearance margin was 0.08 mm higher for males compared to females (*p* < 0.0001); the clearance margin was lower for younger patients compared to patients who were 80 years and older (*p* < 0.0001); and a 0.11 increase in number of MMS stages for high-risk locations compared to low-risk locations (*p* = 0.0008).

Adjusted analyses from the academic center demonstrated that the clearance margin was 0.9 mm higher for males compared to females (*p* < 0.0001); the clearance margin was 0.1 mm higher for every year’s increase in age (*p* < 0.0001) (Table 2); and a 0.25 increase in the number of MMS stages for high-risk pathologic subtypes compared to low-risk pathologic subtypes (*p* < 0.0001) (Table 2). Adjusted analyses from the private practice setting demonstrated that the margin clearance was 0.7 mm higher for males compared to females (*p* = 0.0004); the clearance margin was 0.04 mm higher for every year’s increase in age (*p* < 0.0001) (Table 3); and there was no difference in number of MMS stages for high-risk pathologic subtypes compared to low-risk pathologic subtypes (*p* = 0.07) (Table 2).

## 4. Discussion

Our analyses of this large retrospective study involving 7651 patients with BCC that underwent MMS further improve upon the field’s understanding of MMS practice patterns, in both academic and private practice. Through this high-power study, we determined that a higher clearance margin was associated with increased age and that an increase in the number of MMS stages was associated with higher-risk BCC. By monitoring outliers in MMS practice patterns to identify financial burdens and unnecessary surgery, it may be important to further account for patient demographics as well as tumor factors.

Tumor-specific factors, such as the pathologic subtype, were noted to have a significant impact on practice patterns. While this was not clearly demonstrated in private practice, the results from the academic center showed an association of MMS layers and tumor subtypes with greater MMS layers noted in high-risk BCC subtypes. This is consistent with several studies which have shown a requirement for more MMS layers in high-risk BCC subtypes, such as basosquamous, infiltrative, and morpheaform subtypes, due to higher rates of subclinical spread and recurrence in high-risk subtypes [1,4,5,6,7,8,9,10,11]. Similarly, other studies have shown that primary BCC subtypes, including morpheaform, infiltrative, micronodular, and mixed patterns, were more likely to have positive margins after excision, and, therefore, additional MMS layers were required [12]. Leffel et al. demonstrated that morpheaform or infiltrative BCC subtypes in young adults resulted in larger defect sizes and recurrent rates [12]. Additionally, recurrent BCC tumors of infiltrative subtypes required ≥5 layers to clear the cancer, and, similarly, morpheaform BCC required ≥3 layers to clear the cancer [13].

Furthermore, the subclinical spread of BCC may have influenced the number of MMS stages in our study. We found that there was an increase in MMS stages for high-risk subtypes in the academic setting. Previous studies have shown that high-risk locations may be a risk factor for subclinical spread. Batra et al. demonstrated that body location (including the eyelid, temple, and ear helix), male sex, and age over 35 are significant risk factors for more extensive subclinical spread. These trends were recently supported and further detailed in a larger study, which identified male sex, increased age, Fitzpatrick skin type I, a history of prior BCC, and more aggressive histologic subtypes as correlates to large subclinical extension [8].

While there are no surgeon-specific factors that have been shown to clearly influence MMS practice patterns, several studies have suggested that patient demographics and tumor-specific factors may have an impact [1,4,5,6,7,8,9,10,11,14,15,16,17,18,19,20,21]. In concordance with these studies, our multi-center retrospective study demonstrates that several demographic factors, such as gender and age, as well as a tumor histologic subtype, can influence practice patterns, including clearance size and number of MMS layers when treating BCC. Furthermore, in comparing practice patterns and characteristics of patients of academic center and private practice settings, there were significant differences in demographics, BCC pathologic subtypes, and number of MMS layers.

A patient factor that was found to influence practice patterns was age. Our data from both centers showed a relationship with increasing clearance margins as age increased. It is known that the incidence rates of BCC increase with age and UV radiation exposure. Several studies have also shown larger subclinical spread with older age, which is attributed to increased cumulative sun exposure and skin damage with age; however, the relative contribution of age to the risk of subclinical spread has not been quantified [4,8,20,21]. Consistent with these findings, Dinehart et al. showed that young patients have significantly smaller tumor and defect sizes with no significant differences in tumor location and histologic subtype [22]. In addition to variation in size, similar to the differences noted in gender, a higher cosmetic concern within the younger population has been theorized as a possible reason for the differences in clearance margins or defect size [8]. In addition, it is possible that surgeons may believe that younger patients, as well as female patients, have higher cosmetic concerns than older patients, as well as male patients, respectively.

Both private practice and academic settings did suggest an influence of patient’s gender on practice patterns as evidenced by the statistically significant difference in size of defect between males compared to females. In addition, females had a smaller clearance margin compared to males. Consistent with our findings that a patient’s gender may be associated with differences in practice patterns, several studies have shown differences in practice based on the gender of the patient [2,8,14,15,16,17,18,19]. These differences can be accounted for by several factors, including variation in tumor subtypes, location of tumor, and behavioral practices in males compared to females. While there have been inconsistent data on differences in incidence rates in males compared to females, several studies have shown a higher incidence of BCC in males [8,14,15,16,17,18,19]. The overall higher incidence in males is possibly due to higher levels of lifetime sun exposure in men due to behavioral and lifestyle differences, thus increasing the risk of BCC [8,14,15,16,17]. In addition to variation in incidence, Lee et al. showed differences in tumor pathologic characteristics in males compared to females with more high-risk and invasive subtypes noted in males [23]. One theory for this difference is a stepwise progression of BCC from superficial to nodular to infiltrative subtypes [24]. Given differences in awareness and behavioral practices, including greater adherence to reported self-skin examinations in females, earlier detection may account for the lower risk and superficial subtypes in females [25]. Women were also more likely to have tumors in highly cosmetically sensitive areas such as the central face, including the forehead, nose, and perioral regions, lending itself to the higher cosmetic need for smaller defect sizes [26]. These variations in tumor subtypes and location may further support the smaller clearance margins noted in the females in our study. Another possible theory is that women have a higher overall cosmetic concern in terms of defect size. Although this has not been clearly elucidated, it is supported by the finding that women were more likely to undergo MMS in areas that are often treated by other modalities, such as extremities, due to concern for scar size [7].

Private practices had fewer male patients and performed MMS on more superficial BCC, in addition to more morpheaform/fibrosing/sclerosing/invasive/infiltrative or basosquamous subtypes compared to academic centers. There were also some notable differences in practice patterns, as evidenced by the fewer MMS layers required to achieve clearance in private practice settings. This finding was in contrast to prior studies that have shown outlier practice patterns with a higher number of MMS layers in solo private practice [3]. In this study by Krishnan et al., a higher number of MMS were attributed to differences in payment models, such as volume-specific fee-for-service models in private practices that offer more rewards for higher resections or layers [3].

To aid in determining preoperative margins of nonmelanoma skin cancers (NMSCs), integrating non-invasive imaging techniques, such as dermoscopy, integrated reflective confocal microscopy, and optical coherence tomography, within practice could prove beneficial. However, these techniques rely on the visible contrast between normal and malignant skin, as well as prior clinical experience [27]. Recently, a novel handheld optical polarization imaging (OPI) system was developed for non-invasive, preoperative detection of NMSC margins, which showed promise to positively impact clinical practice. Clinical evaluation of the prototype has shown its superior performance compared to the surgeons’ visual preoperative assessment of lateral cancer margins [28]. This study, conducted by Jermain et al., presented the significant potential of OPI utilization for improving skin cancer treatment outcomes with minimal impact on clinical workflow.

Limitations of this study included the retrospective study design, which analyzed pre-existing data through chart review and was thus subject to confounding due to other risk factors that were not measured, including, for example, behavioral patterns, such as sun exposure, and also patient access to healthcare. Recurrence rates were also difficult to assess, especially if the patient did not return to the same center for follow-up. Furthermore, the results may not be generalizable since only one academic institution and one private practice were included. Also, since the institutions were in different geographic locations, they were not comparable. Despite the limitations, this study had several strengths including a large sample size that spanned several years. The study also included a wide distribution in age, location of tumor, and tumor subtypes. The representation of multiple sites, including one private and one academic center, also allowed us to compare the different types of settings and minimize any confounding that may occur due to differences in practice patterns between the two settings.

## 5. Conclusions

In an era when practice patterns benchmarked to national colleagues are crucial, it is important to consider that patient factors (age, gender) and tumor factors (BCC subtype) may also play an important role in predicting the clearance of the tumor and the required number of MMS stages to achieve it. Importantly, the results of this study may impact clinical practice. For example, based on the results of this study, there may be a difference of 2.9 mm between the clearance margin of a 50-year-old patient and a 79-year-old patient. When trying to conserve as much tissue on some site of the body, 2.9 mm may be clinically significant. When monitoring outlier practice patterns, as well as identifying financial burdens and unnecessary surgery, it is important to account for patient and tumor factors. Further identification and understanding of these characteristics can help optimize patient counseling and preoperative planning.

## Figures and Tables

**Figure 1 cancers-16-02380-f001:**
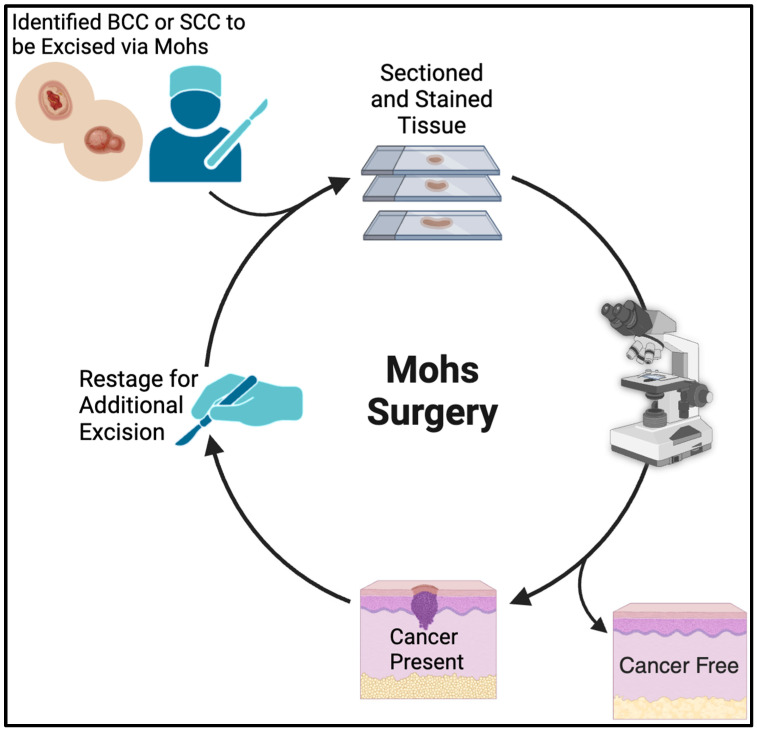
A graphical representation of Mohs Micrographic Surgery (created with https://BioRender.com accessed on 19 June 2024).

**Table 1 cancers-16-02380-t001:** Characteristics of patients, basal cell carcinomas, and Mohs surgeries (academic and private practice).

Variable	Overall	Academic	Private Practice	*p*-Value
	(n = 7651)	(n = 4897)	(n = 2754)	
Age (Median, IQR) years	70 (61, 79)	70 (61, 79)	70 (61, 79)	0.19
Number of Mohs Stages (Median, IQR)	2.0 (1.0, 2.0)	2.0 (1.0, 2.0)	1.0 (1.0, 2.0)	<0.0001
Preoperative Size (Median, IQR) cm	0.8 (0.6, 1.2)	0.8 (0.5, 1.1)	1 (0.7, 1.3)	<0.0001
Postoperative Defect Size (Median, IQR) cm	2 (1.1, 2)	1.5 (1.1, 2.1)	1.4 (1.0, 1.9)	<0.0001
Clearance Margin * (Median, IQR) cm	0.6 (0.3, 0.9)	0.7 (0.5, 1.1)	0.4 (0.2, 0.6)	<0.0001
Recurrence	231 (3.0)	169 (3.5)	62 (2.3)	0.003
Postoperative Subtype				<0.0001
Superficial	300 (4.5)	96 (2.4)	204 (7.7)	
Nodular	3043 (46)	1316 (33)	1727 (65)	
Micronodular	255 (3.8)	151 (3.8)	104 (3.9)	
Morpheaform/Fibrosing/Sclerosing, Invasive/Infiltrative, Basosquamous	550 (8.3)	270 (6.8)	280 (11)	
Superficial and Nodular, Nodular and Micronodular, Micronodular and Nodular and Superficial	1447 (22)	1431 (36)	16 (0.6)	
Nodular and Invasive/Infiltrating, Micronodular and Infiltrative and Superficial	637 (9.6)	612 (15)	25 (0.94)	

* Clearance margin = postoperative defect–preoperative size.

**Table 2 cancers-16-02380-t002:** Academic clearance margins (postoperative defect–preoperative size) and number of Mohs stages, linear regression adjusted model.

Variable, Academic Clearance Margins	Parameter Estimate	*p*-Value
Male (Male vs. female reference)	0.09	<0.0001
Older Age (continuous)	0.01	<0.0001
White (White vs. non-White reference)	0.04	0.67
Tumor Body Location (head/neck vs. trunk/extremities)	−0.18	<0.0001
* Path Subtypes (high- vs. low-risk reference)	0.13	<0.0001
**Variable, Number of Mohs Stages** Male (Male vs. female reference)	−0.03	0.44
Older Age (continuous)	0.01	<0.0001
White (White vs. non-White reference)	0.38	0.04
** Tumor Body Location (high- vs. low-risk reference)	0.25	<0.0001
* Path Subtypes (high- vs. low-risk reference)	0.25	<0.0001

* High-risk path subtypes include morpheaform/fibrosing/sclerosing, invasive/infiltrative, micronodular, basosquamous, or any combination with at least one high-risk path subtype; low-risk subtypes include superficial, nodular, superficial, and nodular. ** High-risk body locations: nose, eyelids, eyebrows, ears, lips, chin, temple, glabella; low-risk body locations: scalp, forehead, cheek, jawline, neck.

**Table 3 cancers-16-02380-t003:** Private practice clearance margins (postoperative defect–preoperative size) and number of Mohs stages, linear regression adjusted model.

Variable, Clearance Margins	Parameter Estimate	*p*-Value
Male (male vs. female reference)	0.07	0.0004
Older Age (continuous)	0.004	<0.0001
Tumor Body Location (head/neck vs. trunk/extremities)	−0.11	<0.0001
* Path Subtypes (high-risk vs. low-risk reference)	0.04	0.06
**Variable, Number of Mohs Stages** Male (male vs. female reference)	0.04	0.29
Older Age (continuous)	0.006	<0.0001
** Tumor Body Location (head/neck vs. trunk/extremities)	0.17	0.0004
* Path Subtypes (high-risk vs. low-risk reference)	0.08	0.07

* High-risk path subtypes include morpheaform/fibrosing/sclerosing, invasive/infiltrative, micronodular, basosquamous, or any combination with at least one high-risk path subtype; low-risk subtypes include superficial, nodular, superficial, and nodular. ** High-risk body locations: nose, eyelids, eyebrows, ears, lips, chin, temple, glabella; low-risk body locations: scalp, forehead, cheek, jawline, neck.

## Data Availability

Redacted data will not be made available as it was derived from patient electronic medical records.

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
