# Peer review of "Factors Influencing Margin Clearance and the Number of Stages of Mohs Micrographic Surgery in Basal Cell Carcinoma: A Retrospective Chart Review"

_cancers, 2024, doi:10.3390/cancers16132380_

Round 1

Reviewer 1 Report

Comments and Suggestions for Authors

This paper, entitled “Factors Influencing Margin Clearance and Number of Stages of Mohs Micrographic Surgery in Basal Cell Carcinoma: A Retrospective Chart Review”, provide information about relationships between clinical and histopathological characteristics and number of MMS stages in patients affected by BCCs.

The topic is interesting. However, some changes are necessary before this work can be considered for publication.

The main criticisms are as follows:

Since Mohs surgery is a complex procedure, a graphic representation of the technique would be useful to better understand it for those who are less experienced.

The retrospective analysis was conducted on two different patient cohorts; the access criteria for MMS and the surgeon's experience in the two settings were similar or not? This point should be clarified, also considering differences reported in the results section. in particular, data relating to different risk factors and different access to  care should be discussed.

Comments on the Quality of English Language

This paper, entitled “Factors Influencing Margin Clearance and Number of Stages of Mohs Micrographic Surgery in Basal Cell Carcinoma: A Retrospective Chart Review”, provide information about relationships between clinical and histopathological characteristics and number of MMS stages in patients affected by BCCs.

The topic is interesting. However, some changes are necessary before this work can be considered for publication.

The main criticisms are as follows:

Since Mohs surgery is a complex procedure, a graphic representation of the technique would be useful to better understand it for those who are less experienced.

The retrospective analysis was conducted on two different patient cohorts; the access criteria for MMS and the surgeon's experience in the two settings were similar or not? This point should be clarified, also considering differences reported in the results section. in particular, data relating to different risk factors and different access to  care should be discussed.

Author Response

Dear Reviewer 1, 

Thank you for reading through our manuscript and providing your constructive feedback on areas for which we can improve our manuscript. We have gone through all your comments and addressed each one within this manuscript update.

  • Since Mohs surgery is a complex procedure, a graphic representation of the technique would be useful to better understand it for those who are less experienced:
    • Line 62-64: We created a graphical representation of the Mohs micrographic surgery technique to aid those less experienced in the field.
  • The retrospective analysis was conducted on two different patient cohorts; the access criteria for MMS and the surgeon's experience in the two settings were similar or not? This point should be clarified, also considering differences reported in the results section. in particular, data relating to different risk factors and different access to care should be discussed:
    • Line 86-88, 108-111: We further discussed that while differences are noted between the academic and private setting, we must look at each practice setting individually to understand how risk factors (patient demographics and tumor factors) influence the practice patterns. We also mention that the Mohs surgeries were performed by board certified Moh surgeons.
    • We also updated all sections to articulate key points with improved clarity.
      • Line 139-144: A brief summary of the results before the discussion.
      • Line 217-226: We included a dedicated paragraph within the discussion that details the role of non-invasive imaging techniques in the definition of pre-operative margins and their implementation in practice.
      • Line 229: We clarify the types of behavioral patterns and risk factors that we were unable to measure due to the nature of the study.
      • Line 245-247: We expanded the age difference in our example, now between a 50 year old and a 79 year old, to demonstrate a larger difference in clearance margin.

We appreciate your time and effort to better this work forward and hope the current manuscript exceeds expectations.

Thank you,

Vincent Azzolino

Reviewer 2 Report

Comments and Suggestions for Authors

Pag 2 line 46: please modify the terms to "low-risk" and "high-risk" BCC

Pag 7 line 240 conclusion: "there may be a difference of 1.9mm between the clearance margin of a 60 year-old patient and 79 year-old patient " please enlarge the age difference as (in example) a 50 year-old patient shows a greater difference of clearer margin compared to a 79 year-old, and the result may be more significative. 

At the beginning of the discussion provide a summary of the results, as then well-analysed alongthe section. In addition add a brief paragraph on the role of non-invasive imaging techniques in the definition of pre-operative margins, and how the use of these devices may impact the clinical practice (i.e. Paradisi et al. Preoperative evaluation of high-risk basal cell carcinoma with line-field confocal optical coherence tomography (LC-OCT) reduces Mohs micrographic surgery stage number: A case-control study; Aleissa S. et al. Presurgical evaluation of basal cell carcinoma using combined reflectance confocal microscopy-optical coherence tomography: A prospective study.

Best Regards

Author Response

Dear Reviewer 2,

Thank you for reading through our manuscript and providing your constructive feedback on areas for which we can improve our manuscript. This was quite helpful, and we have gone through all areas that you have noted.

  • Page 2 Line 46: please modify the terms to "low-risk" and "high-risk" BCC:
    • we have modified the terms to low-risk and high-risk BCC
  • Page 7 Line 240 conclusion: "there may be a difference of 1.9mm between the clearance margin of a 60 year-old patient and 79 year-old patient " please enlarge the age difference as (in example) a 50 year-old patient shows a greater difference of clearer margin compared to a 79 year-old, and the result may be more significative:
    • Line 245-247: Based on your helpful feedback, we increased the age difference in our example, now between a 50 year old and a 79 year old, to demonstrate a larger difference in clearance margin.
  • At the beginning of the discussion provide a summary of the results, as then well-analysed alongthe section:
    • Line 139-145: We provided a brief summary at the beginning of the discussion.
  • In addition add a brief paragraph on the role of non-invasive imaging
  • techniques in the definition of pre-operative margins, and how the use of these devices may impact the clinical practice (i.e. Paradisi et al. Preoperative evaluation of high-risk basal cell carcinoma with line-field confocal optical coherence tomography (LC-OCT) reduces Mohs micrographic surgery stage number: A case-control study; Alissa S. et al. Presurgical evaluation of basal cell carcinoma using combined reflectance confocal microscopy-optical coherence tomography: A prospective study:
    • Line 217-226: We included a dedicated paragraph within the discussion that details the role of non-invasive imaging techniques in the definition of pre-operative margins and their implementation in practice. We appreciate your time and effort to improve this work forward and hope the current manuscript exceeds expectations.

We appreciate your time and effort to better this work forward and hope the current manuscript exceeds expectations.

Thank you,

Vincent Azzolino

Reviewer 3 Report

Comments and Suggestions for Authors Job well done, precisely conceived and carried out with attention to detail. I fully agree with the conclusions of the results and the curiosity to give an explanation to a surgical technique which in my opinion remains to be used only in very selected cases given the costs. I believe you have given the correct indications based on the results

Author Response

Dear Reviewer 3,

Thank you for your comments and reviewing our paper

Sincerely,

Vincent Azzolino

Round 2

Reviewer 1 Report

Comments and Suggestions for Authors

All my prevoius comments have been addressed.

In my opinion the paper can be published in the present form.